# Not getting in too deep: A practical deep learning approach to routine crystallisation image classification

**Jamie Milne** [1,2], **Chen Qian** [2], **David Hargreaves** [2], **Yinhai Wang** [2], **Julie Wilson** [1] *

**1** Department of Mathematics, University of York, York, United Kingdom, **2** AstraZeneca, Cambridge, United Kingdom

☯ These authors contributed equally to this work.

* julie.wilson@york.ac.uk

**Data Availability Statement:** All image files are available from the Dryad repository: https://doi.org/10.5061/dryad.0k6djhb45.

**Funding:** "JM is funded by an EPSRC CASE Studentship, grant number EP/V519807/1, co-

## Abstract

Using a relatively small training set of ˜16 thousand images from macromolecular crystallisation experiments, we compare classification results obtained with four of the most widely-used convolutional deep-learning network architectures that can be implemented without the need for extensive computational resources. We show that the classifiers have different strengths that can be combined to provide an ensemble classifier achieving a classification accuracy comparable to that obtained by a large consortium initiative. We use eight classes to effectively rank the experimental outcomes, thereby providing detailed information that can be used with routine crystallography experiments to automatically identify crystal formation for drug discovery and pave the way for further exploration of the relationship between crystal formation and crystallisation conditions.

## Introduction

The determination of protein structures using X-ray crystallography provides insight into the interaction between a protein and target drug compounds in the drug discovery pipeline. Despite technical advances in the crystallisation process, obtaining suitable crystals for X-ray diffraction is still a major bottleneck in drug design [1]. The current laboratory procedure still relies heavily on trial and error, although crystallisation screens have been developed to sample the search space of different conditions intelligently. Many screens from both commercial and academic sources have been created [2–6], providing combinations of chemicals designed to encourage nucleation and maintain a particular pH. Such screens allow tens of thousands of unique conditions to be tested [7], with the protein construct, concentration, suitable ligand selection and experimental temperature chosen by the crystallographer.

The use of robotics now allows nanolitre volume crystallisation experiments and automated storage and imaging systems are routinely used to record the results. In high-throughput centres, thousands of experiments can be produced each day and each experiment may be imaged multiple times over several weeks. Successful crystallisation rates are thought to be less than 5% [8] and optimization experiments are usually required to obtain X-ray diffraction quality

funded by AstraZeneca as CASE partner. YW, CQ and DH are employed by AstraZeneca UK Ltd and support was provided in the form of salaries for these authors."

**Competing interests:** The authors have declared that no competing interests exist.

crystals from the most positive results in the initial screening. Plate storage and imaging systems can store many crystallisation plates and image the experimental results at pre-specified times whilst maintaining a constant temperature and little disturbance to the experimental plate. Each image time-series can then be extracted for examination, although the monotony of the task can often lead to erroneous classification and observer bias and inter-observer differences are frequent with the human classification of images. An investigation of agreement rates between crystallographers found these were rarely above 70% with this rate decreasing for greater numbers of classes [9]. Visual inspection of images is becoming increasingly impractical and there is an urgent need for automated image analysis.

Automated classification of protein crystallisation images was initially proposed by Zuk & Ward in 1991 when storage and imaging systems were introduced [10]. They used simple Sobel edge detection with the Hough transform to identify straight lines that could indicate the presence of crystals and suggested that more advanced algorithms could be used to monitor crystal growth and better assess the outcome of an experiment. Spraggon et. al [11] (2002) expanded on this idea, using self-organising maps to classify protein crystallisation images as one of six classes, two of which corresponded to crystals, achieving a 75.6% true positive rate for crystals. Since then several research groups have reported improved results using various machine learning algorithms and different numbers of classes. For example, Cumbaa & Jurisica [12] (2005) used linear discriminant analysis to classify protein crystallisation images into four classes namely clear, precipitate, crystal and unknown, whilst Pan and coworkers [13] (2005) classified their images as containing crystals or not using support vector machines. Although this study achieved a false negative rate of less than 3%, this was at the expense of ∼38% false positives. Following the introduction of convolutional neural networks (CNNs) for image analysis [14], Yann & Tang introduced CrystalNet, reporting a 90.8% accuracy with ten classes [15]. However, they used a carefully selected set of cleaned images, for which the class was agreed by multiple crystallographers and the same architecture has since been trained to classify more realistic protein crystallisation images with just 73.7% accuracy for ten classes [16].

The Machine Recognition of Crystallisation Outcomes (MARCO) [17] project in 2018 aimed to develop a classifier that was not specific to a particular imaging system or laboratory and used a training dataset of nearly half a million labelled images from five academic and industrial institutions, including pharmaceutical companies GlaxoSmithKline (GSK), Merck and Bristol Myers Squibb. Although classification rates of >94% were reported for test-set images from multiple imaging platforms, problems with transferability were identified during training. A classifier trained with images from GSK, Merck and the Hauptman-Woodward Institute (HWI) achieved a classification rate of just 61.6% for a test set of images from the Collaborative Crystallisation Centre (C3, CSIRO). However, when C3 images were included in training (making up just 3% of the total training set), 87.5% of C3 images were classified correctly. In fact, transferability is still an issue with the final MARCO classifier [18]. It achieved an accuracy of just 63% for a set of labelled crystallisation images from AstraZeneca (AZ) in our test. Unfortunately, improving the model by transfer learning with representative images from AstraZeneca is not straightforward as it was developed in collaboration with Google scientists and utilised their computational resources. The model and its weights were not published, so the only way to retrain it is to start from scratch. This has motivated the development of a new model that can be trained without the need for supercomputers. Here we compare the most widely used convolutional neural network architectures with an efficient training strategy, which reuses as much as possible pre-trained deep-learning models (freely available from Keras [19]) with initial weights from the Imagenet dataset. We also show that an ensemble classifier obtained by combining these classifiers gives significantly improved

results compared to any of the individual classifiers, achieving an accuracy comparable to the 94% reported in the MARCO project. We provide hyper-parameters that were used with all four networks without further optimisation to allow others to obtain similar results with their own images.

## Data and labelling

The number of classes used to categorise crystallisation images has varied between studies, ranging from just two, *crystal* and *non-crystal*, to as many as ten [20]. If the goal is simply to identify drops containing crystals, then the former might be appropriate but, if the results are to be used in further analyses, for example, to optimise crystallisation conditions and to better understand the crystal formation process, then more classes are required. We have chosen to use the eight classes shown in Table 1 to provide information on the level of success of an experiment. Our categories include three separate crystal classes: *shootable* representing crystals of a quality that could be sent for X-ray diffraction, *optimisable* which includes very small crystals, needles and plates and *crystalline* representing microcrystalline precipitate (often interpreted as "promising looking" by crystallographers). Where initial screens do not result in any crystalline material, information on other experimental outcomes could be used to design optimisation protocols. *Phase separation*, the separation of a protein-rich phase within the crystallisation drop, can provide a boundary where crystals sometimes form later. We consider *light precipitate* and *heavy precipitate* as different outcomes, as light precipitate showing some order indicates supersaturation and can lead to crystal growth whereas heavy amorphous precipitate is unlikely to. The level of precipitate can reveal the suitability of the protein concentration, a fact that has been commercialised by Hampton Research in their Pre-Crystallisation Test (https://hamptonresearch.com). In addition to empty drops, labelled *clear*, we have a *null* class which covers the case where no drop is dispensed as well as experiments affected by factors unrelated to the experiment, such as condensation or issues with focus. Example images from each class are shown in Fig 1.

Our complete dataset consists of 25,316 images from protein crystallisation experiments, labelled according to the classes shown in Table 1. Regardless of the number of classes there will always be disagreements on classification by humans with classes that are not discrete. After a training session led by an experienced crystallographer, a subset of 3095 images was labelled independently by three researchers, allowing the level of agreement to be assessed. Of these images, 2745 were unique with an additional set of 50 duplicated 7 times interspersed. Images were labelled in the same order by each classifier and intra-observer bias measured by the number of times that the 50 repeated images were given the same label. At least 86% of the

**Table 1. Training data.** The eight categories used to classify images in our dataset and the number of images in the training set associated with each class, before and after data augmentation.

| class number | class name | before augmentation | after augmentation |
|---|---|---|---|
| 1 | null | 636 | 4452 |
| 2 | clear | 3589 | 3589 |
| 3 | heavy precipitate | 3493 | 3493 |
| 4 | light precipitate | 4520 | 4520 |
| 5 | phase separation | 668 | 4676 |
| 6 | crystalline | 1834 | 3668 |
| 7 | optimisable | 652 | 4564 |
| 8 | shootable | 925 | 4625 |

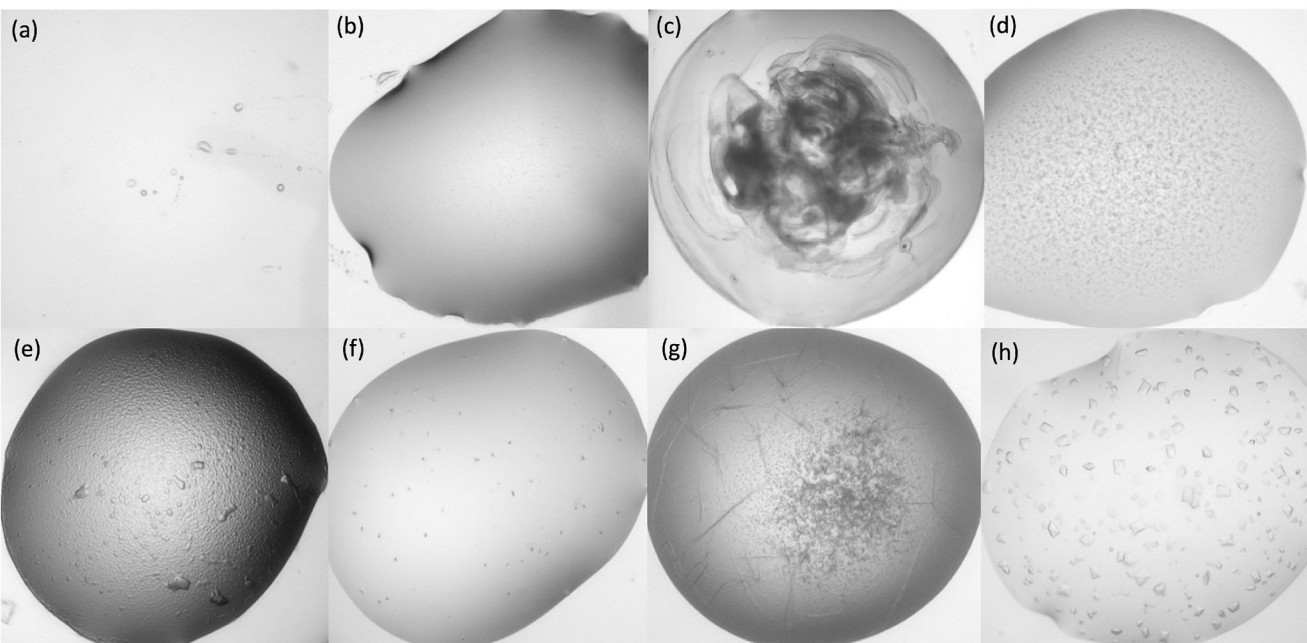

**Fig 1. Example images from each class.** (a) shows a *null* image in which no drop has been dispensed;(b) shows a *clear* drop;(c) shows *heavy precipitate* where the protein is denatured;(d) shows *light precipitate*; (e) shows *phase separation*; (f) shows *crystalline* material; (g) shows *optimisable* crystals and (h) shows *shootable* crystals.

images were given the same label four out of seven times, but scores for more than four agreements dropped drastically with scores for an individual giving the same label all seven times between 10% and 28%. Overall agreement rates between different human classifiers ranged from 50% to 70%, showing the difficulty in classification with continuous outcomes.

Given the low agreement rates between classifiers, a set of 5972 images were given labels agreed by three scientists together to produce an initial training set. The model produced with this training data was then used to classify further images and a simple graphical user interface developed to check the results (S1 Fig in S1 File). Any image for which the classification was considered correct was added to the training set with the assigned label, whereas any image considered misclassified was relabelled before being added. This allowed the size of the training set to increase rapidly but reliably in an iterative procedure that continuously improved the classifier.

As each experiment is imaged multiple times, often with little change between consecutive images, only images from days 1, 8 and 13 were ever included in the training set to reduce redundancy.

Images are mainly from two particular projects, with 7,395 images associated with the receptor tyrosine kinases EGFR [21] and c-KIT [22]. A further 8,922 images are from initial crystallisation screens involving several different proteins to provide the model with a greater variety of experimental outcomes and different crystal forms. In total, 16,317 images were used for training the classifier and Table 1 shows the number in each class.

Classification results were validated with three independent test sets. To assess the effect of including very similar images in the training and test sets, our first test set consists of images from time courses represented in the training data but from different batches (i.e. from days other than 1, 8 and 13). The second test consists of images related to proteins represented in the training set but from different experimental plates and the third test set comprises images

**Table 2. Test data sets.** The number of images in each class for the three different test sets with their similarity to images in the training set.

| | number of images in each class | | similarity to training set |
|---|---|---|---|
| Test set 1 (N = 3000) | Shootable | 131 | Different time point of represented experiment |
| | Optimisable | 296 | |
| | Crystalline | 379 | |
| | Phase separation | 187 | |
| | Light precipitate | 435 | |
| | Heavy precipitate | 859 | |
| | Clear | 418 | |
| | Null | 295 | |
| Test set 2 (N = 3000) | Shootable | 107 | Different experiment from represented project |
| | Optimisable | 135 | |
| | Crystalline | 267 | |
| | Phase separation | 295 | |
| | Light precipitate | 452 | |
| | Heavy precipitate | 1054 | |
| | Clear | 529 | |
| | Null | 161 | |
| Test set 3 (N = 2999) | Shootable | 18 | Unrepresented project |
| | Optimisable | 27 | |
| | Crystalline | 91 | |
| | Phase separation | 364 | |
| | Light precipitate | 598 | |
| | Heavy precipitate | 1275 | |
| | Clear | 614 | |
| | Null | 12 | |

from independent experiments associated with the chemotherapy drug Trastuzumab [23], for which no images were included in the training set. S2 Fig in S1 File shows the distribution of images in each of the three test sets and Table 2 gives the number of images in each.

As colour images multiply the complexity of models and increase computational cost, our RGB images are converted to grayscale, or more specifically luminant (L), where $L = 0.299R + 0.587G + 0.114B$. Each image is cropped to 800x800 pixels, covering the protein crystallisation drop. Class imbalance is a fundamental issue in image classification as unbalanced training data results in models that are biased towards the better represented classes. With an estimated crystallisation rate of <5%, it is especially problematic for automated crystallisation image analysis. Although we limited the number of images in other classes, the three crystal classes (*crystalline*, *optimisable* and *shootable*) account for just 20.9% of the training data with just 5.7% labelled *shootable*. As recognising crystals is a key aim, we incorporated data augmentation into the pre-processing stage to balance the number of images in each training classes. Multiple independent transformations are applied to images from underrepresented classes, these being random rotations, horizontal and vertical flips, and contrast, brightness and resolution adjustments. Each class is duplicated by the rounded inverse of its proportion to the largest class *light precipitate*. The training set is summarised in Table 1.

## Model comparison

The MARCO classifier used an inception-V3 architecture [24] with an additional convolutional layer that reduces the image dimensions from 599x599 to 299x299 pixels. Inception-V3

is a variation on the original Inception module in which a set of convolutional operations occur in parallel before concatenation on output. The V3 module incorporates larger convolutions and regularisation in the form of batch normalisation and label smoothing. This introduces greater complexity but reduces the top-5 error on the ImageNet dataset from 6.67% to 4.2% in comparison to Inception-V1 (GoogLeNet [25]).

Advances in image classification are constantly being made, leading to reduced training times and computational cost as well as improved classification rates. The building blocks, or layers, used to create the architecture of a deep learning network vary and are often combined with new architectures to extend and improve previous ideas. The application programming interface (API) Keras [19] compared network architectures using the ImageNet validation set [26] and provided the results in terms of Top-1 accuracy, where the class given the highest probability is correct, and Top-5 accuracy, which means that one of the five highest probability answers is the correct class. The number of parameters involved and training times are also reported. Computational efficiency in relation to accuracy has been investigated previously for the most popular CNNs [27] and, although NASNet-A-Large was found to have the greatest Top-1 and Top-5 accuracies, it also had the greatest computational cost. Here. we chose to investigate the four architectures with the best-reported Top-1 and Top-5 accuracy that did not exceed our GPU capacity, these being:

1. ResNet50 [28]

2. DenseNet121 [29]

3. InceptionV3 [24]

4. Xception [30]

The architectures for these four networks are given in the (S3-S8 Figs in S1 File). ResNet provides a popular architecture with shortcuts that allow skipped connections to stabilise training but this can compromise the learning capacity of the network. This is mitigated by DenseNet by concatenating all previous feature maps rather than summing them as in ResNet and results in a 2.4% improvement in Top-1 accuracy on the ImageNet validation data with 5.4 million fewer parameters. Similarly, Xception, *aka* 'extreme Inception' was designed to improve on InceptionV3 and achieved a 1.1% increase in accuracy on the ImageNet images while using a million fewer parameters.

Network parameters were optimised using DenseNet121 architecture with a subset of 8,000 images from the MARCO project, split into training (80%) and validation (20%) sets. During optimisation, training was carried out for 20 epochs using a cross-entropy loss function and improvements in validation accuracy were identified as individual parameters were changed. Tests indicated that two additional fully connected layers, separated by dropout layers reduced overfitting and increased validation accuracy. The network parameters found to be optimal (S1 Table in S1 File) were used to train new classifiers with the four chosen architectures and the training data shown in Table 1. A consistent batch size of 16 was used for all architectures regardless of complexity, whilst training occurred for 100 epochs in each case with no early stopping. Weights were optimised using stochastic gradient descent with the Adam optimiser and an initial learning rate of $2e^{-4}$. The learning rate was divided by 2 if the loss had not decreased in 5 epochs. Multiple training runs performed with the DenseNet121 classifier with different subsets of the MARCO dataset as validation data achieved accuracies between 91%—94%.

Transfer learning allows the weights from pre-trained models to be re-used as a starting model for new deep learning problems to save time. However, the models provided for the

ImageNet Large Scale Visual Recognition Challenge (ILSVRC), are based on RGB images and therefore, in order to adapt them to work with our grayscale images, we created two Keras Application models (keras.io/api/applications) for each framework, one accepting RGB images using the ImageNet weights and the other accepting gray scale images with no initial weights specified. The ImageNet weights were transferred to the gray scale model from the layer at which the input shape allowed this. This approach has been shown to improve the efficiency of the model by reducing the number of initial channels without significantly affecting precision [31]. All training was implemented using TensorFlow and Keras and run on four NVIDIA Tesla V100 graphic processing units. The training times per epoch are shown in Table 3.

## Results

Fig 2 shows the performance of the four classifiers in ROC space when considering the three classes, *crystalline*, *optimisable* and *shootable* together as positive results and all other classes as negatives. As expected, the results for all architectures are best for Test set 1, for which images from the same time series were included in the training data. The ResNet50 classifier stands out as most different with noticeably higher sensitivity for Test sets 1 and 2 at the expense of much lower specificity. There are fewer false positives for Test Set 3, comprising images from experiments on a protein not represented in the training data, with this classifier but the sensitivity is also much lower. The DenseNet121 classifier has similar sensitivity for Tests sets 2 and 3 but with lower specificity for Test set 2, where images from experiments with the same protein but not the same time series were included in the training set. Xception gives very similar results to DenseNet for Test set 3, whereas Inception-V3 produces fewer true positives. However, for Test set 2, Inception-V3 shows a slight improvement over DenseNet, whereas a reduction in false positives is outweighed by a reduction in true positives with Xception. It should be noted that the number of images in each class differs between test sets and that Test set 3 in particular has few positive results (see Table 2).

Accuracy is measured as the proportion of classifications that are correct irrespective of class and can therefore be misleading when classes are unbalanced. Cohen's Kappa, originally devised to compare inter-rater agreement, allows for class imbalance when used to assess classification results and so can provide a better representation of performance. Fig 3 shows the Kappa and accuracy scores for the three test sets along with two other common performance metrics, precision and the F1-score. Although the latter two scores are typically used for binary classification, one-vs-all scores can be computed for each class and the weighted average used to provide an overall score. Regardless of the performance metric, it can be seen that Dense-Net121 outperformed the other classifiers on all three test sets. Xception produced the next best results, followed by InceptionV3 and ResNet50 with little to choose between them for Test sets 2 and 3. Results are also shown in S2 Table in S1 File.

It is somewhat surprising that the results obtained for Test set 3, using images from a project not represented in the training data, are consistently better than those for Test set 2, for which images from experiments with same proteins are included. These results are not

**Table 3. Training times per epoch for each classifier.**

| classifier | hours:minutes:seconds |
|---|---|
| DenseNet121 | 0:23:08 |
| ResNet50 | 0:22:52 |
| InceptionV3 | 0:18:45 |
| Xception | 1:16:24 |

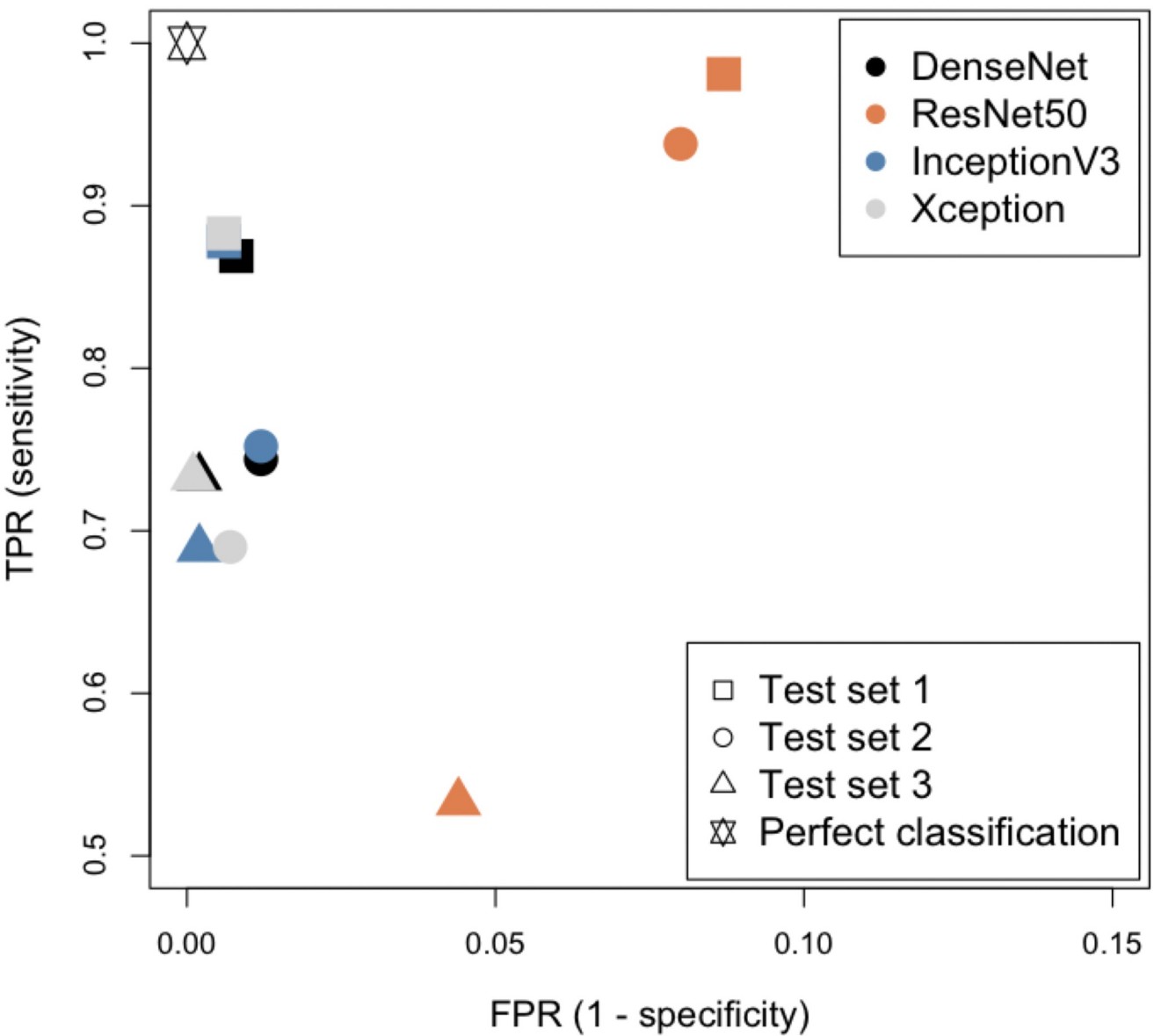

**Fig 2. ROC space comparison of the four chosen architectures with the three different test data sets in Table 2.** Note that only the pertinent area of ROC space is shown.

explained by the fact that there are far fewer examples in the crystal classes. In fact, the results for Test set 3 are proportionally similar to those for Test Set 1 for most classes as can be seen from the full confusion matrices in (S9—S12 Figs in S1 File).

Fig 4 compares the results between test sets obtained with the DenseNet121 classifier. Here balanced accuracy, i.e the average of sensitivity and specificity, is plotted for Test set 1 against each of the other two test sets. For five of the classes, Test set 3 gives very similar results to Test set 1, as can be seen from their proximity to the dotted diagonal line, whereas the results for Test set 2 are lower. Here *heavy precipitate* is not only confused with *light precipitate*, but also *crystalline*. There is also more confusion of *crystalline* with *phase separation* for Test set 2. Test set 2 and 3 results are more similar for the other three classes (*light precipitate, optimisable* and

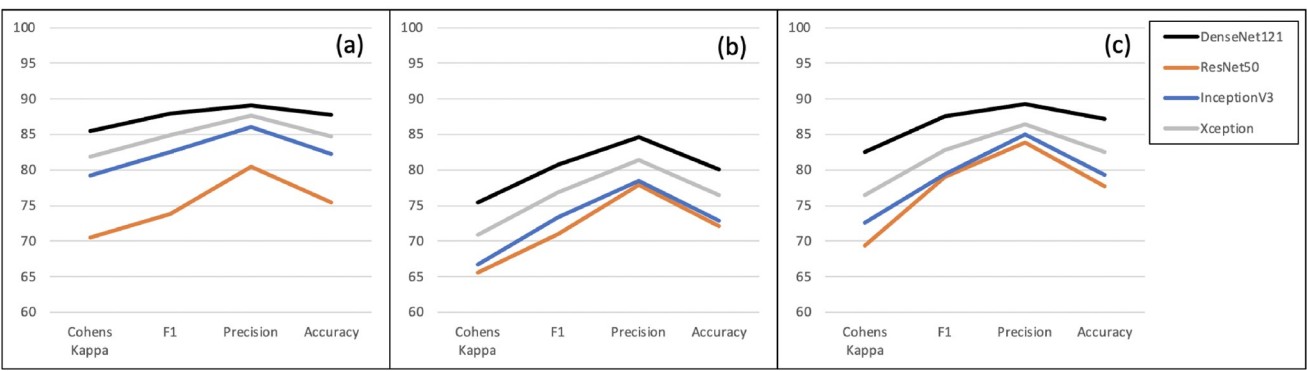

**Fig 3. Performance metrics for the four chosen architectures on (a) Test set 1, (b) Test set 2 and (c) Test set 3.**

*null*) although, apart from *null* which also has few examples in Test set 3, results for Test set 3 are still closer to those for Test set 1. Fig 5 shows the results for Test sets 2 and 3 together, in confusion matrices where the number of classes has been reduced by combining the two precipitate classes and the three crystal classes.

The lowest class accuracy overall is for phase separation with Xception, being confused with both precipitate and crystals. The Xception classifier also confuses both null and clear images with precipitate as does the InceptionV3 classifier. Other regular errors with InceptionV3 are crystals classified as precipitate and phase separation classified as crystals. With ResNet50 the lowest accuracies are for phase separation most often confused with crystal classes and crystals confused with precipitate. As seen earlier, the best results overall are obtained with the DenseNet121 classifier with the lowest accuracy for null images, mainly confused with clear although $\sim 10\%$ of precipitate images are classified as crystals.

The differences in misclassifications between classifiers suggest the possibility of improved classification from ensemble classification. Indeed, majority voting with all four classifiers yields improved balanced accuracies over the DenseNet121 classifier of 13% and 19% for phase separation and crystalline examples respectively with a reduction of just 1 or 2% for *null* and *optimisable* (Table 4). Table 5 shows improvements of up to 21% for overall accuracy and 26% for Cohen's Kappa when the predictions of all four classifiers are combined (Ensemble4) with improvements of 12% for accuracy and 13% for Kappa over the DenseNet121 classifier. We found that choosing the class randomly in the case of tied votes actually gave slightly worse results than simply taking the class that comes up first and realised that this was due to the order that the predictions were provided. The predictions from the DenseNet121 classifier, shown to be the best individual classifier for our data, are given first and so are given priority in the case of ties. Further investigation showed that changing the order of the other three classifiers did not make any difference to the results. However, as the Xception architecture required so much extra training time, we also tried combining the results without this classifier (Ensemble3). Tables 4 and 5 show that correct classifications fall by 3–4%. Comparison of the accuracies in Table 5 with those achieved during training (97–99%) show that, although there is some overfitting of the individual models, this is greatly reduced for the ensemble models and is therefore not a major problem.

## Discussion

The MARCO classifier was trained with over 440 thousand images from five different academic and industrial sources, achieving an accuracy of $\sim 94\%$ on test images from these

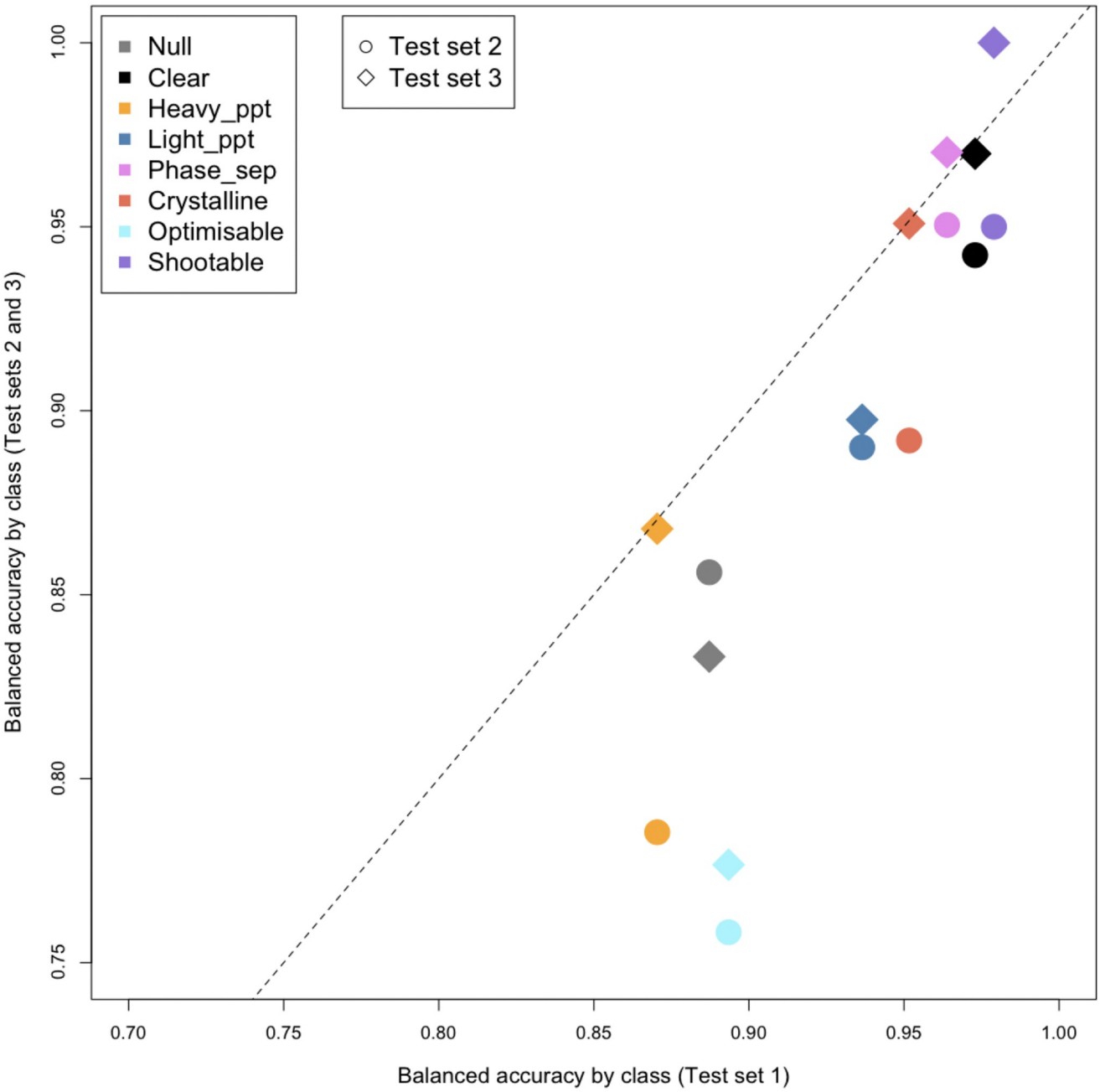

**Fig 4. Comparison of Test set results by class.** For each class, the balanced accuracy obtained for Test Set 1 is plotted against the balanced accuracy for Test Sets 2 and 3, identified by different symbols. The results shown were obtained using the DenseNet121 classifier.

institutions [17]. However, even when the same imaging system is used, subtle differences in the set-up result in much lower accuracies for images from other organisations. Although the classifier is freely available [18] the weights are not, which means that tuning with additional images is not possible and the only way to incorporate new information is to retrain from scratch. Rosa and colleagues characterised the image data used in the original MARCO model investigated the training settings most likely to enhance the local performance of a classification model based on these images [32]. However, we found that using only our own images

| DenseNet 121 | Predicted class | | | | |
|---|---|---|---|---|---|
| True class | null | clear | ppt | phase-sep | crystal |
| null | 75.4 | 11.2 | 4.6 | 2.9 | 5.9 |
| clear | 0.1 | 92.6 | 0.6 | 0.5 | 6.1 |
| ppt | 0.3 | 0.8 | 84.9 | 4.2 | 9.8 |
| phase-sep | 0.0 | 0.0 | 1.0 | 94.6 | 4.4 |
| crystal | 0.1 | 0.1 | 1.7 | 3.4 | 94.8 |

| ResNet 50 | Predicted class | | | | |
|---|---|---|---|---|---|
| True class | null | clear | ppt | phase-sep | crystal |
| null | 86.8 | 2.6 | 3.9 | 0.9 | 5.7 |
| clear | 7.7 | 90.9 | 0.2 | 0.4 | 0.7 |
| ppt | 2.2 | 3.8 | 86.3 | 4.7 | 3.0 |
| phase-sep | 0.8 | 2.5 | 2.5 | 79.5 | 14.7 |
| crystal | 1.0 | 3.0 | 11.0 | 4.9 | 80.1 |

| Inception V3 | Predicted class | | | | |
|---|---|---|---|---|---|
| True class | null | clear | ppt | phase-sep | crystal |
| null | 87.6 | 0.2 | 11.8 | 0.2 | 0.7 |
| clear | 5.1 | 76.8 | 12.6 | 2.0 | 3.6 |
| ppt | 0.4 | 0.1 | 95.8 | 1.0 | 2.8 |
| phase-sep | 0.6 | 0.0 | 5.8 | 83.8 | 9.8 |
| crystal | 0.1 | 0.1 | 9.7 | 2.0 | 88.1 |

| Xception | Predicted class | | | | |
|---|---|---|---|---|---|
| True class | null | clear | ppt | phase-sep | crystal |
| null | 82.2 | 3.5 | 11.4 | 0.0 | 2.9 |
| clear | 1.8 | 84.0 | 11.3 | 0.3 | 2.6 |
| ppt | 0.3 | 0.1 | 93.7 | 0.2 | 5.8 |
| phase-sep | 0.2 | 0.2 | 13.1 | 65.8 | 20.8 |
| crystal | 0.3 | 0.4 | 3.5 | 0.2 | 95.6 |

**Fig 5. Confusion matrices showing the results obtained with the four different classifiers for Test sets 2 and 3 combined.** The number of classes has been reduced for easier comparison by combining *light precipitate* and *heavy precipitate* as just *precipitate* and the three classes *crystalline, optimisable* and *shootable* as *crystal*.

gave better results than including even subsets of the MARCO images. Our training set is very small in comparison but diverse and carefully labelled and we have shown that similar results are achieved for independent test data whether from projects represented in the training data or not.

We compared four of the most widely-used network architectures, chosen by their performance reported by the Keras API but taking into account computational limitations. It is possible that with greater computational capacity more advanced models, such as inception-v4 and SENet-154, could further improve classification rates as they do with the ImageNet validation data. Each of our chosen architectures has been used in recent studies on the detection of

**Table 4. Balanced class accuracies for Tests sets 2 and 3 combined obtained using ensemble classification in comparison to those obtained for individual classifiers.** Ensemble4 combines the results of all four classifiers, whereas ensemble3 does not use the predictions from the Xception classifier. Accuracies greater than 90% are highlighted.

| classifier | null | clear | heavy ppt | light ppt | phase sep | crystalline | optimisable | shootable |
|---|---|---|---|---|---|---|---|---|
| DenseNet121 | 0.97 | 0.95 | 0.93 | 0.93 | 0.86 | 0.79 | 0.92 | 0.94 |
| ResNet50 | 0.86 | 0.91 | 0.87 | 0.88 | 0.82 | 0.89 | 0.71 | 0.99 |
| InceptionV3 | 0.93 | 0.98 | 0.77 | 0.87 | 0.91 | 0.85 | 0.96 | 0.84 |
| Xception | 0.96 | 0.97 | 0.81 | 0.89 | 0.97 | 0.81 | 0.97 | 0.88 |
| Ensemble4 | 0.96 | 0.99 | 0.97 | 0.98 | 0.99 | 0.98 | 0.90 | 1.00 |
| Ensemble3 | 0.96 | 0.98 | 0.94 | 0.95 | 0.98 | 0.95 | 0.91 | 97.0 |

**Table 5. Overall accuracy and Kappa values for ensemble classification in comparison to those obtained from individual classifiers.** Ensemble4 combines the results of all four classifiers, whereas ensemble3 does not use the predictions from the Xception classifier.

| classifier | Accuracy | Cohen's Kappa |
|---|---|---|
| DenseNet121 | 0.83 | 0.81 |
| ResNet50 | 0.74 | 0.68 |
| InceptionV3 | 0.78 | 0.73 |
| Xception | 0.81 | 0.88 |
| Ensemble4 | 0.95 | 0.94 |
| Ensemble3 | 0.92 | 0.90 |

COVID-19 from chest X-Ray images, sometimes in combination. Rahimzadeh and Attar concatenated the output features from Xception and ResNet50V2 to provide a classification model that could differentiate between COVID-19 and pneumonia [33], achieving an overall accuracy of 91.4%. With many more normal and pneumonia cases, they had just 30 COVID-19 cases in the test set for which the sensitivity was 75.3%. A balanced dataset comprising 320 chest X-Ray images from each of COVID-19 patients and healthy volunteers was used in the ratio 50:20:30 for training, validation and testing [34]. Zhang and colleagues compared the results of three DenseNet architectures (121,169 and 201) with different transfer learning settings. They found that DenseNet201 gave the best results with an overall accuracy of 96.9% and suggested that its deeper neural structure could learn more complicated patterns. A comparison of three other architectures (ResNet50, InceptionV3 and VGG16) was carried out by Guefrechi and colleagues with a larger data set of chest X-Ray images [35]. The full data set consisted of 623 COVID-19 positive images augmented to 2000 together with 3000 COVID-19 negative images. They fine-tuned each classifier and obtained similar results for the different architectures, with accuracies of 97.1% to 98.3% on test data.

Previous benchmark analysis of different architectures found that Xception gave the highest accuracy, followed by InceptionV3, ResNet50 and then DenseNet121 [27], whereas, for our data, the DenseNet121 architecture outperformed the other architectures. However, we did obtain very similar accuracy with Xception and, in fact, did give a higher Kappa score for Test sets 2 and 3 combined. Although the ResNet50 architecture produced the worst predictions overall, this classifier had a high crystal detection rate of 93.8%. It does however produce more false positives as can be seen in Fig 2. The training time for the Xception classifier was four times that for InceptionV3, requiring 127 hours over the 100 epochs in comparison to between 31 and 39 hours for the other three classifiers. As the four classifiers had different strengths in terms of particular class accuracies, we were able to combine their results in an ensemble classifier using majority voting and achieve accuracy comparable to that seen in the MARCO project. We consider the increase in accuracy over the best individual classifier to be worth the combined training time.

We have chosen to categorise the results of crystallisation experiments using eight classes rather than the four used in the MARCO projects in order to provide useful information for further analysis. Whereas the MARCO classes combined both good and bad results in the ambiguously named *other* class, we aim to show just how good an outcome is by having more classes. With the exception of *null*, which could represent unavoidable technical problem rather than an experimental outcome, our eight classes reflect some ranking, even within the three crystal classes, that could inform further work. We have developed a graphical user interface (GUI) to visualise the results for each experimental plate and Fig 6 shows an example obtained using the Morpheus crystallisation screen [4]. This screen was designed as a grid with

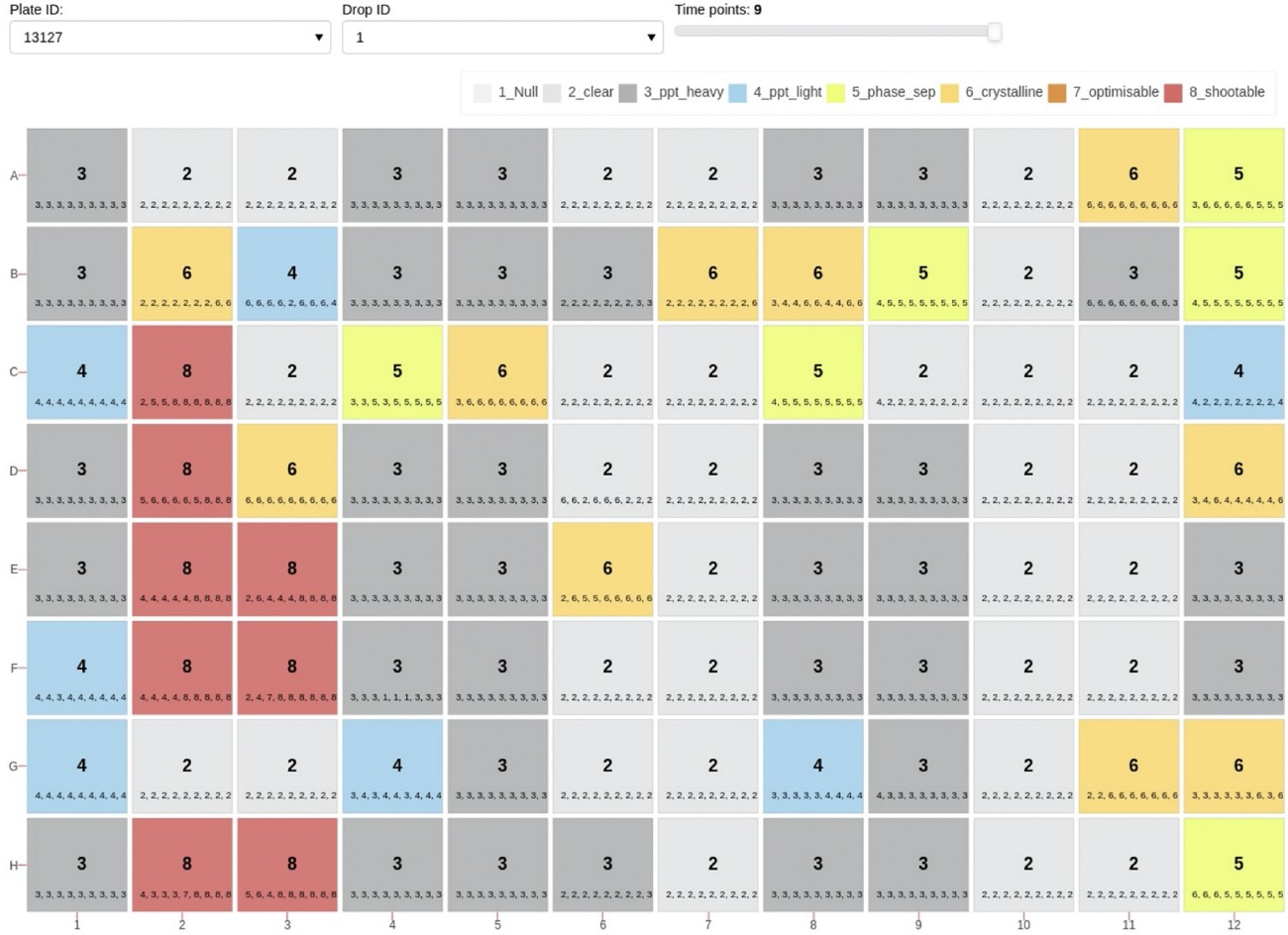

**Fig 6. Visualisation of experimental outcomes obtained using the Morpheus crystallisation screen [4].** In addition to being labelled with the predicted class number, each well in the plate is colour-coded according to the predicted class. The sequence of numbers in small text for each well shows the class numbers predicted earlier in the time series.

columns covering 4 different precipitant-cryoprotectant mixes for three different buffer systems (pH 6.5, 7.5 and 8.5) and rows corresponding to different additives. Patterns can be seen in Fig 6 with certain columns being associated with particular classes, while some additives are noticeable for being different, for example row B (halides) and row G (carboxylic acids). As with the MARCO Polo interface, developed to display and share the results of the MARCO classifier [36], our GUI allows images in the time course to be displayed and provides experimental details. Patterns such as those seen in Fig 6 suggest that successful conditions could potentially be related to particular protein properties, allowing optimal conditions for new proteins to be predicted.

## Conclusion

We have shown that a bespoke ensemble classifier can be trained to classify images from crystallisation experiments without the need for supercomputers or a huge training data set. Comparison of four popular deep learning architectures revealed interesting differences in class accuracies between these classifiers. While DenseNet121 provided the best overall accuracy, each architecture was optimal for at least one particular class. The results in Table 4 show that

DenseNet121 actually only gave the best results for *null* images and the two precipitate classes, whereas InceptionV3 gave the highest accuracy for it clear images, Xception was best for *phase separation* and *optimisable* images and ResNet50 produced the best results for images in the *crystalline* and *shootable* classes. Although the overall accuracy achieved with ResNet50 is 9% lower than that for DenseNet121, this classifier did in fact have an accuracy for *crystalline* images 10% higher than that with the DenseNet architecture. These differences in misclassification suggested ensemble classification could improve the results and the use of majority voting with all four classifiers increased the overall accuracy from 83%, obtained for the best individual classifier, to 95% and Cohen's Kappa from 81% to 94%.

## Supporting information

**S1 File. Supplementary information file consisting of supplementary figures and tables.** (PDF)

## Acknowledgments

We would like to thank scientists at AstraZeneca UK Ltd for providing data and feedback.

## Author Contributions

**Conceptualization:** David Hargreaves, Yinhai Wang, Julie Wilson.

**Data curation:** Chen Qian.

**Formal analysis:** Jamie Milne, Chen Qian, Julie Wilson.

**Funding acquisition:** David Hargreaves, Yinhai Wang.

**Investigation:** Jamie Milne, Chen Qian.

**Methodology:** Chen Qian, Julie Wilson.

**Project administration:** David Hargreaves, Yinhai Wang.

**Resources:** Chen Qian, David Hargreaves, Yinhai Wang.

**Software:** Jamie Milne, Chen Qian.

**Supervision:** Chen Qian, David Hargreaves, Yinhai Wang, Julie Wilson.

**Validation:** Chen Qian, Julie Wilson.

**Visualization:** Jamie Milne, Julie Wilson.

**Writing – original draft:** Jamie Milne, Julie Wilson.

**Writing – review & editing:** Jamie Milne, Chen Qian, David Hargreaves, Yinhai Wang, Julie Wilson.

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
