## [Decision Letter · Decision Letter 0]

23 Dec 2022

PONE-D-22-27534Not getting in too deep: a practical deep learning approach to routine crystallisation image classification

PLOS ONE

Dear Dr. Wilson,

Thank you for submitting your manuscript to PLOS ONE. After careful consideration, we feel that it has merit but does not fully meet PLOS ONE’s publication criteria as it currently stands. Therefore, we invite you to submit a revised version of the manuscript that addresses the points raised during the review process.

Please revise your manuscript according to reviewer 1's comments in your revised manuscript.

We look forward to receiving your revised manuscript.

Kind regards,

Yan Chai Hum

Academic Editor

PLOS ONE

Journal Requirements:

"This work was possible thanks to the award of an EPSRC CASE studentship to JM (reference EP/V519807/1). This was co-funded by AstraZeneca UK Ltd as industrial CASE partner. YW, CQ and DH are employed by AstraZeneca UK Ltd and support was also provided in the form of salaries for these authors. We would also like to thank scientists at AstraZeneca UK Ltd for providing data and feedback."

"JM is funded by an EPSRC CASE Studentship, grant number EP/V519807/1, co-funded by AstraZeneca as CASE partner. https://www.ukri.org/councils/epsrc/

YES- YW, CQ and DH are employed by AstraZeneca UK Ltd and support was provided in the form of salaries for these authors. " ext-link-type="uri" xlink:type="simple">https://www.astrazeneca.com/"

Additional Editor Comments (if provided):

Please revise according to reviewer's comment.

Reviewers' comments:

Reviewer's Responses to Questions

**Comments to the Author**

1. Is the manuscript technically sound, and do the data support the conclusions?

Reviewer #1: Partly

Reviewer #2: Yes

2. Has the statistical analysis been performed appropriately and rigorously? 

Reviewer #1: No

Reviewer #2: Yes

3. Have the authors made all data underlying the findings in their manuscript fully available?

Reviewer #1: No

Reviewer #2: Yes

4. Is the manuscript presented in an intelligible fashion and written in standard English?

Reviewer #1: Yes

Reviewer #2: Yes

5. Review Comments to the Author

Reviewer #1: Dear Authors, revise the paper as per following comments:

authors should provide the comments of the cited papers after introducing each relevant work. What readers require is, by convinced literature review, to understand the clear thinking/consideration why the proposed approach can reach more convinced results. This is the very contribution from authors. In addition, authors also should provide more sufficient critical literature review to indicate the drawbacks of existed approaches, then, well define the main stream of research direction, how did those previous studies perform? Employ which methodologies? Which problem still requires to be solved? Why is the proposed approach suitable to be used to solve the critical problem? We need more convinced literature reviews to indicate clearly the state-of-the-art development.

As far as I know, the values of hyper-parameter (including the number of hidden units in the deep neural network) will greatly affect the performance of the model. At the same time, the value of the hyper-parameter is related to the specific data and model, and it is not easy to be determined. Even with the same data set, different models may have different optimal hyper-parameter. However, the experiments in this manuscript did not explain the determination of the values for the hyper-parameter. Please add experiments or description to select the optimal hyper-parameter.

More experiments and some comparisons with other up-to-date methods should be addressed or added to back your claims to expand your experiments and analysis of results further.

Better to add a little more large-scale experiment in your revision if possible.

Which feature selection method did you use for feature extraction? How did you come to know the important features and skipped the rest ones? Please mention the features used for your experiment in Table and give solid reason for the selection of those features.

Can you please explain when using different datasets what is the impacts on model training process? How did you use it for your research work? Please explain.

The authors made a passing reference to how they prevent overfitting their model. (Some more explanation is necessary to support the techniques selected (random disturbing and rotating) over other approaches)

The training loss function is a function that quantifies the difference between the predicted value and the actual value throughout the training process. It is critical to design a fair and effective loss function for the target model's training; which technique is utilized as the loss function? [discussion is required]

Please provide some comments on the novelty of the proposed methodology. How is it different from other methodology such as ResNet50 and DenseNet121? How is it more efficient than those networks?

Experimental uncertainty should be highlighted

Reviewer #2: good writing, easy to follow, and the presented works seems reproducible, the reported methodology are technically sound. References are adequate and up to date. However, the architecture of the 4 common used neural networks are not explained in their manuscript.

6. PLOS authors have the option to publish the peer review history of their article (what does this mean?). If published, this will include your full peer review and any attached files.

Reviewer #1: No

Reviewer #2: No

---

## [Author Response · Author response to Decision Letter 0]

1 Feb 2023

Please see responses below each point raised by the reviewers. In addition we have uploaded all images to the Dryad repository (https://doi.org/10.5061/dryad.0k6djhb450, corrected formatting errors and removed funding information from the manuscript as requested by the editor.

Reviewer #1: Dear Authors, revise the paper as per following comments:

authors should provide the comments of the cited papers after introducing each relevant work. What readers require is, by convinced literature review, to understand the clear thinking/consideration why the proposed approach can reach more convinced results. This is the very contribution from authors. In addition, authors also should provide more sufficient critical literature review to indicate the drawbacks of existed approaches, then, well define the main stream of research direction, how did those previous studies perform? Employ which methodologies? Which problem still requires to be solved? Why is the proposed approach suitable to be used to solve the critical problem? We need more convinced literature reviews to indicate clearly the state-of-the-art development.

Whilst we also reference earlier relevant work, we believe that the use of convolutional neural networks in the MARCO paper (Bruno et al., PLOS one 13(6), 2018) represents the current state of the art in protein crystallisation image classification and the aim here is to address the main issue with the MARCO classifier, that is, the problem of transferability and the fact that retraining with images representative of those to be classified is required. As retraining with the original MARCO architecture is not possible without super-computers, we chose to compare the most popular convolutional network architectures that can be used without the need for extensive computational resources and have shown that the same level of accuracy can be achieved. We have changed the end of the introduction to make our aims clearer. This now reads:

“Here we compare the most widely used convolutional neural network architectures with an efficient training strategy, which reuses as much as possible pre-trained deep-learning models (freely available from Keras [19]) with initial weights from the Imagenet dataset. We also show that an ensemble classifier obtained by combining these classifiers gives significantly improved results compared to any of the individual classifiers, achieving an accuracy comparable to the ~94% reported in the MARCO project. We provide hyper-parameters that were used with all four networks without further optimisation to allow others to obtain similar results with their own images.”

As far as I know, the values of hyper-parameter (including the number of hidden units in the deep neural network) will greatly affect the performance of the model. At the same time, the value of the hyper-parameter is related to the specific data and model, and it is not easy to be determined. Even with the same data set, different models may have different optimal hyper-parameter. However, the experiments in this manuscript did not explain the determination of the values for the hyper-parameter. Please add experiments or description to select the optimal hyper-parameter.

In order that readers may repeat our approach with their own images, we carry out an efficient training strategy, which reuses as much as possible the pre-trained deep-learning models with initial weights from the Imagenet dataset. Therefore hyper-parameters such as the number of hidden units in the four specific network architectures we have chosen to compare and combine are fixed rather than optimised. We now provide details on the four network architectures (ResNet50, DenseNet121, InceptionV3 and Xception) in the Supplementary Information.

Other important hyperparameters which are independent of the architectures, such as epochs and learning rates, were optimised for the DenseNet121 model using data from the MARCO project rather than the data from our own project. We used these parameters with all four networks in our study to show that as we aimed to provide parameters that readers could use and show that these can give results comparable to the MARCO classifier without the need for time-consuming hyperparameter optimisation procedures. The parameters used are given in Supplementary Table S1 and described in the paper as follows: 

“A consistent batch size of 16 was used for all architectures regardless of complexity, whilst training occurred for 100 epochs in each case with no early stopping. Weights were optimised using stochastic gradient descent with the Adam optimiser and an initial learning rate of 2e−4. The learning rate was divided by 2 if the loss had not decreased in 5 epochs.”

More experiments and some comparisons with other up-to-date methods should be addressed or added to back your claims to expand your experiments and analysis of results further. Better to add a little more large-scale experiment in your revision if possible.

As stated above, we aimed to compare our results with the accuracy reported for the MARCO classifier (one of the largest scale and advanced study in the field of deep learning in crystallography) without the need for supercomputers. We are able to achieve accuracy for our data comparable to the ~94% reported in the MARCO project and note that the MARCO classifier was only 63% accurate on our data. With over 16,000 images used to train the classifiers and ~9,000 images as independent test data, we believe our dataset is large enough and particularly wanted to show that a huge training data set was not required in order to achieve accuracy comparable to the MARCO classifier. Transferability is a major issue and we aimed to show that equally good results could be achieved using standard convolutional network architectures and provide a roadmap to allow readers to implement their own classification system. 

Which feature selection method did you use for feature extraction? How did you come to know the important features and skipped the rest ones? Please mention the features used for your experiment in Table and give solid reason for the selection of those features.

As described in the paper, we did manually crop the original images to the central region as only the changes in this area (the crystallisation drop within the well) provides information on the crystallisation experiment. There is no manual feature selection process as might be used with classical machine learning methods as the process of feature selection is an integral part of convolutional neural networks. The input 800X800 pixel image can be seen as 6400 initial input features which, after all convolution layers, the model automatically condenses to much fewer features (128 in this study) which the classification is based on. 

Can you please explain when using different datasets what is the impacts on model training process? How did you use it for your research work? Please explain.

During training to optimise parameters, multiple runs were performed with the Densenet121 classifier for which different subsets of the MARCO dataset were chosen as validation data, and accuracies between 91% - 94% were achieved. We have now added this information to the paper as follows: 

“Multiple training runs performed with the Densenet121 classifier with different subsets of the MARCO dataset as validation data achieved accuracies between 91% - 94%.”

Furthermore, cross-validation results obtained during training for the four different architectures ranged from 97.3% to 99.5%. This shows that results for different datasets differ by 3%. 

The authors made a passing reference to how they prevent overfitting their model. (Some more explanation is necessary to support the techniques selected (random disturbing and rotating) over other approaches)

We mentioned overfitting in the context of training, stating that “two additional fully connected layers, separated by dropout layers reduced overfitting and increased validation accuracy.” Overfitting cannot be prevented, only detected by comparison of the results obtained for the training set (for which the model is optimised) with those on validation or test data (which has not been used to build the model). During training we used cross-validation to indicate overfitting and chose our training strategy according to results on validation data, adding drop-out layers and stopping training earlier when the validation score plateaued. 

However, once our models are fully-trained, we evaluate the results on three entirely independent test sets. These data have never been seen or used in training and therefore the models cannot be biased towards them. The test sets are large enough not to be affected by chance (i.e. just being lucky in the choice of test set) and comparison with the training set accuracy shows that the models are not overfitted. 

We have also added the following to the end of the Results section:

“Comparison of the accuracies in 5 with those achieved during training (97-99%) show that, although there is some overfitting of the individual models, this is greatly reduced for the ensemble models and is therefore not a major problem.”

Rather than the cross-validation method, it could be that the reviewer is referring to the disturbing of individual images. Image augmentation is a popular method which alters original images by adding a range of small image transformations to provide pseudo-new images. As the performance of deep learning neural networks can improve with the amount of data available, image augmentation is often applied to create additional training images from existing ones. Augmentation is an effective approach to data balancing by oversampling images from under-represented classes and prevent overfitting of the larger classes. We described the augmentation approach in the paper as:

Multiple independent transformations are applied to images from underrepresented classes, these being random rotations, horizontal and vertical flips, and contrast, brightness and resolution adjustments.

Although we now say “these being” rather than “including” as this suggested not all transformations were listed.

The training loss function is a function that quantifies the difference between the predicted value and the actual value throughout the training process. It is critical to design a fair and effective loss function for the target model's training; which technique is utilized as the loss function? [discussion is required]

We stated that the loss function used was the cross-entropy loss function. For clarity, we have now added the definition for the cross-entropy function to the Supplementary Information.

Please provide some comments on the novelty of the proposed methodology. How is it different from other methodology such as ResNet50 and DenseNet121? How is it more efficient than those networks?

We provide a comparison of the most widely used convolutional neural networks that can be used without the need for supercomputers, including Densenet121 and Resnet50. We also show that an ensemble classifier obtained by combining these classifiers gives significantly improved results compared to the use of any individual classifier, achieving an accuracy comparable to the ~94% reported in the MARCO project. We believe that our conclusions reflect our contributions:

“We have shown that a bespoke ensemble classifier can be trained to classify images from crystallisation experiments without the need for supercomputers or a huge training data set. Comparison of four popular deep learning architectures revealed interesting differences in class accuracies between these classifiers. While DenseNet121 provided the best overall accuracy, each architecture was optimal for at least one particular class. ” 

and

“These differences in misclassification suggested ensemble classification could improve the results and the use of majority voting with all four classifiers increased the overall accuracy from 83%, obtained for the best individual classifier, to 95% and Cohen's Kappa from 81% to 94%.”

Experimental uncertainty should be highlighted

As mentioned above, we have now added information on the variance during training, from multiple runs with the Densenet121 classifier and different subsets of the MARCO dataset (accuracies between 91% - 94%) and during training with the four different architectures (97.3% - 99.5%). 

Reviewer #2: good writing, easy to follow, and the presented works seems reproducible, the reported methodology are technically sound. References are adequate and up to date. However, the architecture of the 4 common used neural networks are not explained in their manuscript.

We thank the reviewer for their comments. We now provide details on the four network architectures (ResNet50, DenseNet121, InceptionV3 and Xception) in the Supplementary Information.

---

## [Editor Report · Decision Letter 1]

21 Feb 2023

Not getting in too deep: a practical deep learning approach to routine crystallisation image classification

PONE-D-22-27534R1

Dear Dr. Wilson,

We’re pleased to inform you that your manuscript has been judged scientifically suitable for publication and will be formally accepted for publication once it meets all outstanding technical requirements.

Kind regards,

Yan Chai Hum

Academic Editor

PLOS ONE
---

## [Editor Report · Acceptance letter]

27 Feb 2023

PONE-D-22-27534R1 

Not getting in too deep: a practical deep learning approach to routine crystallisation image classification 

Dear Dr. Wilson:

I'm pleased to inform you that your manuscript has been deemed suitable for publication in PLOS ONE. Congratulations! Your manuscript is now with our production department. 

Kind regards, 

on behalf of

Dr. Yan Chai Hum 

Academic Editor

PLOS ONE